# Antibacterial and Immunostimulatory Activity of Potential Probiotic Lactic Acid Bacteria Isolated from Ethiopian Fermented Dairy Products

Seyoum Gizachew [1,2], Wannes Van Beeck [2], Irina Spacova [2], Max Dekeukeleire [2], Ashenafi Alemu [3], Wude Mihret Woldemedhin [3], Solomon H. Mariam [4], Sarah Lebeer [2,*] and Ephrem Engidawork [1,*,†]

1   Department of Pharmacology and Clinical Pharmacy, School of Pharmacy, College of Health Sciences, Addis Ababa University, Addis Ababa P.O. Box 9086, Ethiopia
2   Department of Bioscience Engineering, Faculty of Sciences, University of Antwerp, 2020 Antwerp, Belgium
3   Armauer Hansen Research Institute, Addis Ababa P.O. Box 1005, Ethiopia
4   Aklilu Lemma Institute of Pathobiology, Addis Ababa University, Addis Ababa P.O. Box 1176, Ethiopia
*   Correspondence: sarah.lebeer@uantwerpen.be (S.L.); ephrem.engidawork@aau.edu.et (E.E.)
†   Hold senior authorship.

**Abstract:** Lactic acid bacteria (LAB) form a group of bacteria to which most probiotics belong and are commonly found in fermented dairy products. Fermented foods and beverages are foods made through desired microbial growth and enzymatic conversions of food components. In this study, 43 LAB were isolated from Ethiopian traditional cottage cheese, cheese, and yogurt and evaluated for their functional and safety properties as candidate probiotics. Twenty-seven isolates, representative of each fermented food type, were selected and identified to the species level. *Limosilactobacillus fermentum* was found to be the predominant species in all samples studied (70.4%), while 11.1% of isolates were identified as *Lactiplantibacillus plantarum.* All 27 isolates tested showed resistance to 0.5% bile salt, while 26 strains were resistant to pH 3. The LAB isolates were also evaluated for antagonistic properties against key pathogens, with strain-specific features observed for their antimicrobial activity. Five strains from cottage cheese (*Lactiplantibacillus plantarum* 54B, 54C, and 55A, *Lactiplantibacillus pentosus* 55B, and *Pediococcus pentosaceus* 95E) showed inhibitory activity against indicator pathogens that are key causes of gastrointestinal infections in Ethiopia, i.e., *Escherichia coli*, *Salmonella enterica* subsp. *enterica* var. Typhimurium, *Staphylococcus aureus*, *Shigella flexneri*, and *Listeria monocytogenes*. Strain-specific immunomodulatory activity monitored as nuclear factor kappa B (NF-κB) and interferon regulatory factor (IRF) activation was documented for *Lactiplantibacillus plantarum* 54B, 55A and *P. pentosaceus* 95E. Antibiotic susceptibility testing confirmed that all LAB isolates were safe concerning their antibiotic resistance profiles. Five isolates (especially *Lactiplantibacillus plantarum* 54B, 54C, and 55A, *Lactiplantibacillus pentosus* 55B, and *P. pentosaceus* 95E) showed promising results in all assays and are novel probiotic candidates of interest for clinical trial follow-up.

**Keywords:** traditional fermented dairy products; lactic acid bacteria; antimicrobial activity; NF-κB; interferon regulatory factors; probiotics; Ethiopia

## 1. Introduction

Food fermentation forms an essential element of human civilization, serving as a means to preserve and enhance shelf-life, flavor, texture, taste, nutritional value, and functional properties of food [1,2]. Fermented foods and beverages are foods made through desired microbial growth and enzymatic conversions of food components [3]. Africa is considered to be a continent with the richest variety of fermented foods [4]. Especially, Ethiopia is a country rich in cultural diversity, with each cultural group having its own variety of fermented food and beverages [5]. Fermented food items commonly consumed in Ethiopia include fermented dairy products (e.g., cottage cheese (Ayib), yogurt (Ergo)),

fermented plants (e.g., Enjerra, Kotcho), fermented beverages (e.g., Borde, Cheka), and fermented condiments (e.g., Siljo, Awaze, Datta) [5–7]. Most of these traditional fermented foods are produced on a fairly small-scale level, usually for household consumption and, at times, sold by local vendors from their homes [5,6]. However, their microbiology and potential health benefits are not yet widely studied. Moreover, there is a rapid rise in the number of industrially processed fermented products in urban areas, especially dairy products [8].

Most fermented dairy products harbor a microbial community characterized by a dominance of lactic acid bacteria (LAB) that can ferment carbohydrates to produce lactic acid. This group of bacteria includes several genera, such as the emended genus *Lactobacillus* [9], *Lactiplantibacillus*, *Lacticaseibacillus*, *Limosilactobacillus*, *Streptococcus*, *Pediococcus*, *Leuconostoc*, and *Weissella* [1,9]. Because of their long-time use in various food and feed preparations without pronounced adverse effects, many species of LAB (especially those belonging to *Lactobacillaceae*) have been granted a "generally recognized as safe" (GRAS) status by the US FDA [10] and "Qualified Presumption of Safety" (QPS) by the European Food Safety Authority (EFSA) [11]. Probiotics are defined as "live microorganisms that, when administered in adequate amounts, confer a health benefit to the host" [12]. According to this definition, the health benefit must be supported by at least one positive human clinical trial conducted according to generally accepted scientific standards [13]. Over the last decades, LAB use as probiotics has increased because specific LAB strains can confer a wide range of health benefits through mechanisms including enhancement of gut barrier function, competitive exclusion of pathogens, production of antimicrobial substances [14], and modulation of immune functioning [15]. These mechanisms of action can result in clinical benefits such as those documented for specific strains in specific clinical trials, especially for reducing the risk or symptoms of various gastrointestinal (GI) disorders such as irritable bowel syndrome, ulcerative colitis, and bacterial or viral infections [16].

Foodborne bacterial and viral infections are an important cause of morbidity and mortality and a significant barrier to the socio-economic development of all nations. In 2010, based on a World Health Organization (WHO) estimation, Africa was reported to have the highest burden of foodborne diseases per capita, with a median of 2455 foodborne Disability Adjusted Life Years (DALYs) per 100,000 inhabitants [17]. Of these, 26.6% were attributed to *Salmonella* spp., 11.2% to enteropathogenic *Escherichia coli*, 8.6% to enterotoxigenic *E. coli*, 0.08% to *Listeria monocytogenes*, 5.7% to *Campylobacter* spp., and 0.004% to Shiga-toxin producing *E. coli* [18,19]. In Ethiopia, diarrheal diseases have been reported to be the second most important contributor to the total burden of all disease types and the second leading cause of premature death [19]. Meta-analyses on the burden of methicillin-resistant *Staphylococcus aureus* (MRSA) and *Shigella* species in Ethiopia provided a pooled prevalence of 32.5% [20] and 6.6% [21], respectively. Antibiotic resistance has also increased worldwide, posing an enormous clinical and public health burden, necessitating the search for alternatives to deal with the emerging risk of resistant pathogens [22]. Probiotics could form a valuable approach to decrease the burden of foodborne diseases in a cost-efficient manner because they can target different steps in the infection processes through multifactorial modes of action [23].

One mode of action of probiotics, and especially LAB, is their capacity to directly inhibit the growth of bacterial, fungal, and even viral pathogens via their capacity to produce the broad-acting antimicrobial molecule lactic acid and more species- or strain-specific antimicrobials such as bacteriocins [24]. Another key mode of action of probiotics is modulation of the mucosal immune system, whereby probiotics can activate the host cells to produce antimicrobial molecules or cellular activities [25,26]. This activity is generally mediated via microbe-associated molecular patterns (MAMPs) expressed by the probiotics, which can interact with various immune receptors on the host cells, such as Toll-like receptors [27]. This interaction leads to activation of nuclear transcription factors such as NF-κB that play a key signaling role in induction of immune responses following a variety of stimuli, such as with MAMPs [28,29]. While NF-κB induces a number of genes

mainly involved in pro-inflammatory cascades at sites of infection to kill pathogens, the intestinal epithelium generally does not trigger inflammatory responses against commensal bacteria but induces tolerance towards the commensal microorganisms. However, some of the signals induced by commensals and probiotics could result in better alertness and more rapid clearance of incoming pathogens. Another important signaling pathway in response to microbial stimuli is related to interferon (IFN) production, which is regulated by interferon-regulatory factors (IRFs) [30,31]. This pathway is necessary for efficient antiviral responses and is generally induced by viral MAMPs [30]. Spacova et al. [32] also found that several selected strains of probiotic lactobacilli can induce this pathway and boost antiviral responses. However, this mechanism has not been widely explored for LAB isolated from traditional fermented foods.

Most probiotic strains are selected without a detailed investigation of the underlying modes of action. Thus, there is a high demand for new strains with specific therapeutic modalities against infectious and other diseases [14]. In this study, we aimed to mine the microbial diversity of fermented foods and beverage items in Ethiopia for novel potential probiotic strains. Interesting isolates were characterized and evaluated for specific antimicrobial and immunological properties.

## 2. Materials and Method

### 2.1. Isolation and Characterization of LAB Strains

One yogurt and one cheese product from two different dairy industries in Addis Ababa, Ethiopia, and two traditional cottage cheeses from the Arba Minch district in Ethiopia were aseptically collected. The process of fermentation used to produce traditional cottage cheeses is spontaneous and uncontrolled. To isolate LAB, 10 mL (g) of each sample was suspended and homogenized in 90 mL phosphate-buffered saline (PBS) (pH 7−7.4). The homogenized sample (1st dilution) was used to prepare ten-fold serial dilutions, and 10 μL of the appropriate dilution (mostly the 3rd to 6th) was spread-plated on de Man, Rogosa, and Sharpe (MRS) agar (Hi-Media, Mumbai, India), a selective medium used to enrich LAB [33]. These plates were then incubated anaerobically (BD BBL™ GasPak™ jars) at 37 °C for 24 to 48 h. Plates with 30 to 300 colonies were selected, and colonies were counted. Five colonies were then randomly selected based on their differing appearance and purified through three successive streaking on MRS agar, in which aliquots of the selected isolates were stored at −80 °C in MRS broth containing 25% glycerol. Finally, the pure isolates were characterized presumptively as LAB by cell morphology, Gram staining, catalase test, and motility according to standard procedures [1], whereby Gram-positive, catalase-negative, and non-motile isolates were presumptively identified as LAB. The number of colony-forming units per milliliter/gram (CFU/mL(g)) was calculated as a function of the number of confirmed LAB colonies and the inoculated dilution using the following formula [34]:

$$\text{CFU/mL} = \text{total colonies present} \times \text{percent confirmed colonies} \times \text{dilution.} \tag{1}$$

### 2.2. Molecular Identification of LAB Isolates

The selected isolates presumptively identified as LAB (Gram-positive, catalase-negative, and non-motile) were further identified through 16S rRNA gene sequencing. For the detection of LAB strains using 16S rRNA gene sequences, the following primers were used: 27F (5′-AGAGTTTGATCCTGGCTCAG-3′) and 1492R (5′-GGTTACCTTGTTA CGACTT-3′). The bacterial genomic DNA was extracted using a 16S rRNA gene colony PCR technique. In brief, a colony was picked, mixed up, and vortexed in 10 μL molecular grade water. The cells were lysed through microwaving for 2 × 1.5 min at 800 W. The master mix was prepared in a clean room and contained 2.5 μL 10×VWR Buffer, 0.5 μL dNTPs (10 mM), 2.5 μL 27F (10 μM), 2.5 μL 1492R (10 μM), 0.2 μL Taq polymerase, and 6.8 μL molecular grade water to make a master mix of 15 μL final volume for each sample. This 15 μL master mix was then added to each tube containing a 10 μL DNA template. PCR was

performed under the following conditions: initial activation at 95 °C for 2 min; denaturation step cycles at 95 °C for 30 s; annealing step at 55 °C for 30 s; extension step at 72 °C for 1 min and 30 s; and final extension cycle at 72 °C for 5 min; for 30 cycles. A total of 5 μL of the PCR product was used to run 1% agarose gel electrophoresis on a gel with 5 μL GelRed dye. Successful samples (bright band at 1500 bps) were sent for sequencing (Sanger sequencing at Neuromics Support Facility VIB, Uantwerpen). The resulting sequences were analyzed using SeqTrace 0.9.0 software and submitted to a search for similarity in the EzBioCloud.net 16S-based ID. Bacterial species identification was assumed when the query sequence showed pairwise similarity of >98.7% for the 16S rRNA gene sequence, as previously described [35].

*2.3. Resistance of LAB Isolates to Gastrointestinal Conditions In Vitro*

LAB isolates from overnight (18 h) cultures (in MRS broth at 37 °C) were harvested (4000× *g*, 10 min, 4 °C), washed twice with PBS, and adjusted to $1.5 \times 10^8$ CFU/mL. To determine survival of the LAB strains in acidic conditions mimicking the GI tract, 100 μL of $1.5 \times 10^8$ CFU/mL of each LAB strain was added to 900 μL of sterile PBS adjusted to pH 3.0 (using 1M HCl) and then incubated under stirring (150 rpm) at 37 °C for 3 h, mimicking the time spent by food in the stomach. After incubation, 50 μL of each bacterial solution was collected, and 10-fold serial dilutions were prepared using sterile PBS and spread plated onto MRS agar in triplicates for enumeration of viable cells. To determine survival of the LAB strains in bile salt solution, 100 μL of $1.5 \times 10^8$ CFU/mL was added into 900 μL of sterile PBS (pH 8.0) supplemented with 0.5% (*w*/*v*) bile salts. The bacterial solution was then incubated at 37 °C under stirring (150 rpm) for 4 h, mimicking the time spent by food in the small intestine [36–38]. The percentage (%) of cell survival was calculated as shown below:

$$\% \text{ cell survival} = (\log \text{CFU}_T / \log \text{CFU}_C) \times 100$$

where $\text{CFU}_C$ and $\text{CFU}_T$ represent the total viable count of LAB isolates before and after, respectively, incubated under the simulated GI condition (low pH or bile salts). The starting absolute number was $1.5 \times 10^8$ CFU/mL, and the experiment's limit of detection was $10^3$ CFU/mL.

*2.4. Antagonistic Activity of LAB Isolates against Indicator Pathogens*

Antagonistic activity of the LAB isolates against the foodborne pathogens was evaluated via spot overlay and radial diffusion assays with *Salmonella* spp., *Shigella* spp., *Escherichia coli*, *Listeria* spp., and *Staphylococcus* spp. as indicators of antimicrobial activity. In addition, a longitudinal liquid culture growth assay was performed using *S. aureus* MI/1310/1938.

2.4.1. Spot Overlay Assay

This assay was performed at both Armauer Hansen Research Institute (AHRI), Addis Ababa, Ethiopia, and the Laboratory of Applied Microbiology and Biotechnology (LAMB), University of Antwerp, Antwerp, Belgium. The indicator pathogenic bacteria used in AHRI were *S. aureus* (ATCC 25923), *L. monocytogenes* (ATCC 19115), and *E. coli* (ATCC 25922) obtained from the Ethiopian Public Health Institute, and a clinical isolate of MRSA obtained from Tikur Anbessa Specialized Hospital, Addis Ababa University, Ethiopia. At the LAMB, *L. monocytogenes* MB2022 isolated from Wijnendaele cheese, *S. enterica* subsp. *Enterica* var. Typhimurium NTCT 13347, *E. coli* O157:H7 BRMSID188 lacking pathogenicity *stx* genes (for biosafety reasons) isolated from bovine [39], *S. aureus* MI/1310/1938—methicillin-sensitive (MSSA), and *S. flexneri* LMG 10472 were used as indicator strains. For the spot overlay assay, 2 μL from each LAB isolate, cultivated overnight (20–24 h) in MRS broth under micro-aerobiosis, was spotted on the surface of agar media (AHRI: MRS agar for all pathogens tested; LAMB: Mueller Hinton agar (MHA) (1.5%) supplemented with 5 g/L glucose for *S. aureus* and LB agar (1.5%) supplemented with 5 g/L glucose for other pathogens) as described previously [38,40]. After spotting, the plates were incubated aerobically at

37 °C for 24 h (for LAB spots on MRS agar) and 48 h (for LAB spots on MHA and LB agar). A volume of overnight growth of each indicator pathogen required to make a final concentration of $5 \times 10^6$ CFU/mL was mixed with 20 mL of soft agar (0.5% agar) and uniformly poured over the spot inoculated square plate (7 mL/round Petri dish). The plates were then incubated aerobically at 37 °C for 24 h. The antagonistic activity was recorded as the diameter (mm) of zone of inhibition. A total of 2 μL of hexetidine (0.1%) or chlorhexidine 0.2% were spotted as positive controls, while MRS broth was spotted as negative control. Experiments were run in triplicates and the average values were recorded.

### 2.4.2. Radial Diffusion Assay

This assay was performed as described elsewhere [40] using the same indicator pathogens, media, and final concentration of the pathogen inoculum as mentioned in the spot overlay assay performed at LAMB. LAB strains were first cultivated overnight (20–24 h) in MRS broth micro-aerobically (non-shaken) at 37 °C. The supernatants of these cultures (ca. $10^9$ cfu/mL) were collected through centrifugation (at $2484 \times g$, 15 min, 4 °C) and filter sterilized with a 0.22 μm filter, with or without pH adjustment to pH 7.4. An adequate volume of overnight growth of indicator pathogens was added to cooled agar (55 °C) and mixed well to produce a final concentration of $5 \times 10^6$ CFU/mL and poured onto a square plate. LAB cell-free culture supernatants (CFS) (45 μL), pH adjusted (7.4) or non-adjusted, were dispensed into 6 mm diameter wells drilled using a sterile glass Pasteur pipette. The plates were aerobically incubated at 37 °C for 24 h. After incubation, antagonistic activity was recorded as the diameter (mm) of growth inhibition zones around each well. In this assay, MRS broth (45 μL) and hexetidine (0.1%, 45 μL) were used as negative and positive controls, respectively. Experiments were run in triplicates and the average values were recorded.

### 2.4.3. Antimicrobial Activity Screening of Cell-Free Culture Supernatants in Liquid Culture Assays

This assay was also performed as described previously [40]. Briefly, 190 μL of a diluted overnight (20–24 h) culture of *S. aureus* MI/1310/1938 (ca. $10^5$ cfu/mL) was added to the wells of a microplate supplemented with 10 μL CFS of LAB strains (obtained in the same way as in the radial diffusion assay) to obtain a total volume of 200 μL. A total of 10 μL 0.1% hexetidine and 10 μL MRS and LB medium were used as positive and negative control, respectively. Bacteria were grown, and optical density (OD) was measured at 600 nm ($OD_{600}$) each 30 min for 24 h using a Synergy HTX multi-mode reader. Each test was measured in triplicates, and the average $OD_{600}$ was calculated.

### 2.5. Assessment of Immunostimulatory Activity of LAB Isolates

Immunostimulatory activity of the LAB strains was assessed by measuring activation of the NF-κB pathway and IRF pathway in human THP1-Dual™ reporter monocytes (InvivoGen, San Diego, CA, USA), as previously described [32]. The cells were maintained according to the manufacturer's instructions in growth medium containing RPMI 1640, 2 mM L-glutamine, 25 mM HEPES, 10% heat-inactivated fetal bovine serum, 100 μg/mL Normocin™ and Pen-Strep (100 U/mL;100 μg/mL). The bacterial cells were UV-inactivated in a biosafety level 2 cabinet for 90 min with vortexing after each 15 min before co-incubation with THP1-Dual™ cells. In the immunostimulation assay, UV-inactivated bacterial cells (final concentration $10^7$ CFU/mL before inactivation) were added to THP1-Dual™ cells (final concentration $10^6$ cells/mL) and co-incubated for 24 h at 37 °C and 5% $CO_2$. For assessment of the NF-κB pathway activation, secreted embryonic alkaline phosphatase (SEAP) activity in the THP1-Dual™ monocyte supernatant after addition of a p-nitrophenyl phosphate (pNPP) solution was measured (absorbance) at 405 nm according to the manufacturer's instructions. IRF pathway induction was measured by assessing the activity of a secreted luciferase (Lucia) by using QUANTI-Luc buffer, a luciferase detection reagent,

based on luminescence using a BioTek Synergy HTX multi-mode reader according to the manufacturer's instructions.

### 2.6. Antibacterial Susceptibility Testing of LAB Isolates

Antibacterial susceptibility of selected LAB strains was determined for ampicillin, chloramphenicol, clindamycin, erythromycin, gentamycin, kanamycin, streptomycin, and tetracycline as per the recommendations of EFSA [41], using a broth microdilution test previously described [42], with minor modifications. In brief, 10 µL of each antibacterial solution was dispensed into each well of a 96-well microplate containing 180 µL of MRS broth. Subsequently, a 10 µL-culture aliquot of each test LAB isolate was added to each well (final viable cell count of approximately 7 log CFU/mL). The microplates were sealed with plastic bags to prevent bacterial dehydration. The experiments included controls, in particular bacteria alone, MRS broth, and known probiotic control strains, *Lacticaseibacillus rhamnosus* GG [43] and *Lactiplantibacillus plantarum* WCFS1 [44], and were performed in triplicates. The system was then aerobically and statically incubated at 37 °C for 48 h, and the plates were observed for any visible growth. The strains that showed visible growth were considered resistant.

### 2.7. Statistical Analysis

Results are expressed as mean ± standard deviation. Normal distribution of data was evaluated using Shapiro–Wilk and Kolmogorov–Smirnov normality tests before statistical comparisons. For normally distributed data, one-way ANOVA followed by Dunnett's multiple comparisons test was employed. Otherwise, the Kruskal–Wallis's test, followed by Dunn's multiple comparisons test, was used. Statistical comparisons were made when applicable using GraphPad Prism version 9.2.0. Differences were considered statistically significant at $p < 0.05$.

## 3. Results

In this study, samples were taken from a representative fermented yogurt and a typical cheese obtained from different large-scale commercial dairy farms in Addis Ababa, Ethiopia. In addition, two representative traditionally fermented cottage cheese samples from the Arba Minch district, Ethiopia, were taken. The traditionally fermented cottage cheese samples were prepared in a similar traditional method at the household level by heating a fermented (18–24 h) and defatted cow milk. The commercial cheese sample used in this study was a type of soft cheese produced from pasteurized milk coagulated by adding a starter culture and rennet, whereas the yogurt sample was produced commercially by fermenting pasteurized cultured milk. Following anaerobic cultivation on MRS agar, 54 microbial isolates were obtained; 43 were identified putatively as LAB based on morphological characteristics because they were Gram-positive bacilli or cocci, catalase-negative and non-motile. Of these 43 isolates, 27 were selected based on the degree of antibacterial activity displayed (16 showed poor activity, see Supplemental Table S2) and to cover diversity of the sample origins. The samples were then subjected to a screening pipeline to select potential probiotic strains, as depicted in Figure 1. The LAB load of the dairy samples in CFU/mL(g) is presented in Supplemental Table S1. The data revealed that more CFU were obtained than LAB from traditionally fermented products than industrially fermented products.

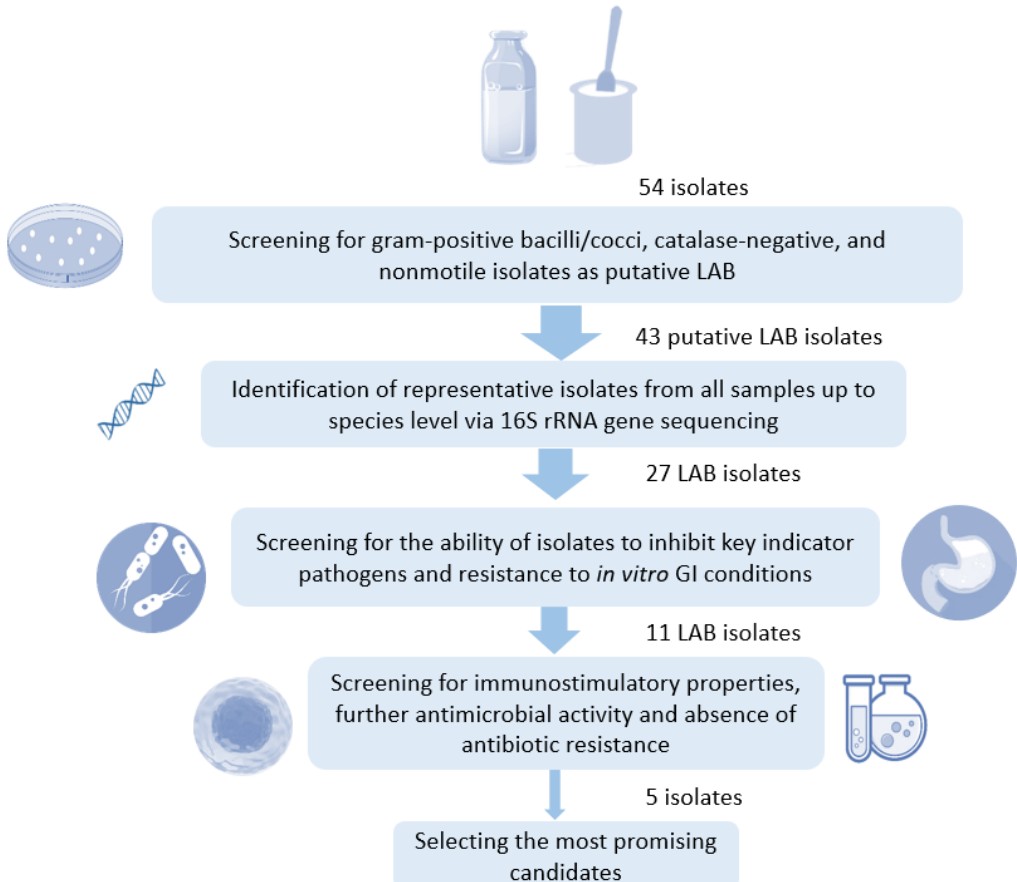

**Figure 1.** Strain selection flow chart used to select potential probiotic strains from Ethiopian yogurt and cheese products based on a combination of phenotypic and genotypic methods. Fifty-four isolates were obtained from Ethiopian yogurt and cheese-based products. Of these, 43 were classified as putative LAB. After initial antimicrobial analysis, 27 isolates were selected (taking into account origin) for 16S rRNA analysis. Of these 27 isolates, 11 were selected based on initial antimicrobial screening and species diversity for in-depth characterization of their probiotic potential. Of these, 5 isolates scored the best on all tests and were selected as the most promising probiotic candidates.

### 3.1. Selected LAB Isolates from Ethiopian Fermented Dairy Products Predominantly Belong to the Genus Limosilactobacillus

The 27 selected isolates were identified up to species level with 16S rRNA gene Sanger sequencing (Table 1). *Limosilactobacillus fermentum* showed to be the predominant species (19/27; 70.4%) identified, while 11.1% of isolates were identified as *Lactiplantibacillus plantarum*. Eleven of the twelve selected LAB isolates from the yogurt sample were identified as *Limosilactobacillus fermentum*. Seven of the fourteen selected isolates from spontaneously fermented cottage cheese samples were also identified as *Limosilactobacillus fermentum*, while the remaining isolates were identified as probably *Lactiplantibacillus plantarum* (three isolates), *Weissella confusa* (93A), *Pediococcus pentosaceus* (95E), *Lactiplantibacillus pentosus* (55B), and *Enterococcus lactis* (54A). The query sequence showed that the pairwise similarity of all strains was >99.7% for the 16S rRNA gene sequence of the top hits.

**Table 1.** *16S rRNA*-gene-based identification of LAB isolates from Ethiopian dairy products.

| Source | Strain | Identified by 16S rRNA as: | Pairwise Similarity (%) | Selected (Yes) |
|---|---|---|---|---|
| Commercially fermented yogurt | 12A | *Limosilactobacillus fermentum* | 99.92 | Yes |
| | 12D | *Limosilactobacillus fermentum* | 100 | |
| | 12E | *Limosilactobacillus fermentum* | 100 | |
| | 13A | *Limosilactobacillus fermentum* | 99.91 | |
| | 13C | *Limosilactobacillus fermentum* | 100 | |
| | 13E | *Limosilactobacillus fermentum* | 100 | |
| | 14C | *Limosilactobacillus fermentum* | 100 | |
| | 14D | *Limosilactobacillus fermentum* | 100 | |
| | 15B | *Limosilactobacillus fermentum* | 100 | |
| | 15C | *Limosilactobacillus fermentum* | 100 | |
| | 15D | *Limosilactobacillus fermentum* | 100 | |
| | 15E | *Streptococcus thermophilus* | 99.92 | Yes |
| Commercially fermented cheese | 25A | *Limosilactobacillus fermentum* | 99.92 | Yes |
| Spontaneously fermented cheese | 54A | *Enterococcus lactis* | 99.77 | |
| | 54B | *Lactiplantibacillusplantarum* | 100 | Yes |
| | 54C | *Lactiplantibacillusplantarum* | 100 | Yes |
| | 55A | *Lactiplantibacillus plantarum* | 100 | Yes |
| | 55B | *Lactiplantibacillus pentosus* | 100 | Yes |
| | 55E | *Limosilactobacillus fermentum* | 100 | Yes |
| | 93A | *Weissella confusa* | 100 | Yes |
| | 93B | *Limosilactobacillus fermentum* | 99.92 | |
| | 93E | *Limosilactobacillus fermentum* | 99.85 | |
| | 94C | *Limosilactobacillus fermentum* | 99.85 | |
| | 94D | *Limosilactobacillus fermentum* | 99.84 | |
| | 94E | *Limosilactobacillus fermentum* | 99.84 | Yes |
| | 95A | *Limosilactobacillus fermentum* | 99.85 | |
| | 95E | *Pediococcus pentosaceus* | 100 | Yes |

*3.2. Selected Isolates Show High In Vitro GI Resistance*

In order to act as a probiotic in the GI tract and exert their beneficial effect on the host, the ingested LAB must survive the acidic conditions in the stomach and resist bile acids in the small intestine. Therefore, the survival of the selected LAB isolates was investigated in simplified stomach- and bile-mimicking conditions using a starting absolute number of $1.5 \times 10^8$ CFU/mL (Figure 2). All 27 LAB isolates tested showed resistance to 0.5% bile salt, with 15 LAB isolates having viability of more than 80% after 4 h exposure. Exposure to low pH (pH = 3) for 3 h, simulating the time spent by food in the stomach, revealed that 26 of the 27 LAB isolates exhibited resistance. Overall, the LAB isolates tested showed better tolerance capacity to 0.5% bile salt exposure than to low pH.

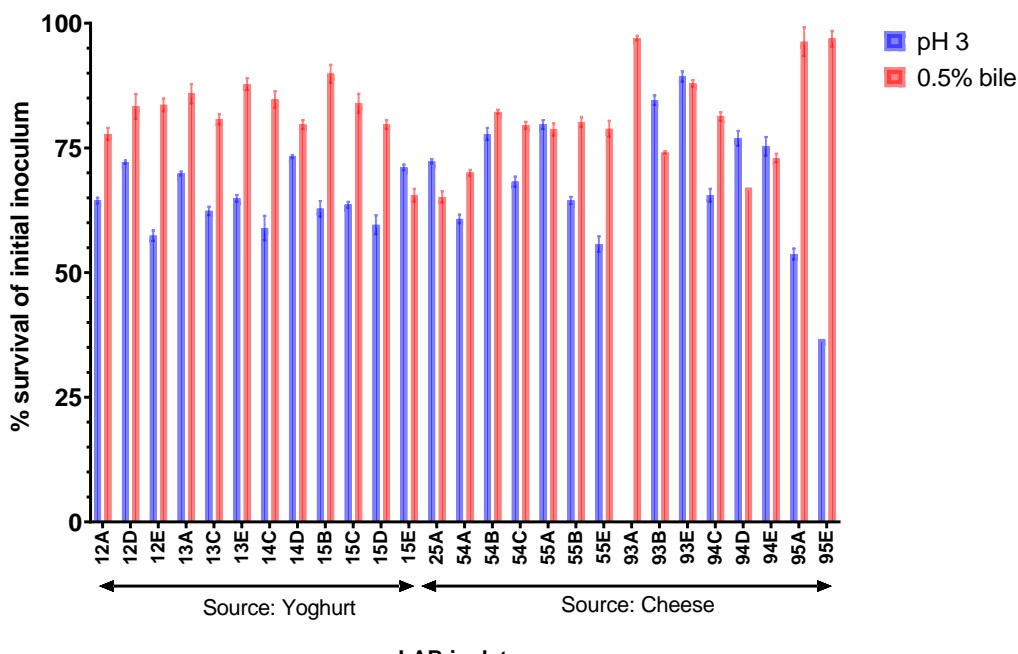

**Figure 2.** Percentage of survival (from initial inoculum) of the selected LAB isolates after exposure to acidic pH and bile salt solution. Isolates were exposed to pH 3.0 for 3 h at 37 °C and 0.5% (*w/v*) bile salt solution (pH 8.0) for 4 h at 37 °C under stirring (150 rpm). Data are expressed as mean ± SD per condition (*n* = 3).

### 3.3. LAB Isolates from Ethiopian Fermented Foods Inhibit Indicator Foodborne Pathogens

Antagonistic activity of the 27 selected isolates was evaluated against four indicator pathogens, i.e., *L. monocytogenes* ATCC 19115, *S. aureus* ATCC 25923, *E. coli* ATCC 25922, and a clinical MRSA via spot overlay assay. The 27 isolates tested were found to inhibit these pathogens at varying degrees (Supplemental Table S2). A total of 18 LAB isolates (5 from yogurt, 12 from cottage cheese) displayed inhibition activities against all the pathogens tested to a varied extent (Supplemental Table S3). A total of 18 of the 27 isolates tested also showed a wider inhibition zone against indicator pathogens compared to the positive control (chlorhexidine 0.2%). However, eight of the LAB isolates (all from yogurt) failed to show activity against MRSA.

Subsequently, based on the spot assay results and species variety, 11 isolates were selected for more detailed characterization. First, more detailed profiling of their antimicrobial activity was performed against more pathogens using both radial diffusion and spot overlay assays (Table 2). In a radial diffusion assay, the activity of the secreted LAB metabolites was studied, while a spot assay investigated the activity of the live LAB. Five indicator pathogens, including pathogens that are among the key causes of GI infections in Ethiopia (*S. enterica* subsp. *enterica* Typhimurium, *E. coli* O157:H7 (-*stx* genes), *S. aureus* MI/1310/1938, MSSA, *S. flexneri* LMG 10472, and *L. monocytogenes* MB2022) were studied. Six (*L. plantarum* 54B, 54C, 55A, *Lactiplantibacillus pentosus* 55B, *W. confusa* 93A, and *P. pentosaceus* 95E) of the eleven LAB strains tested were effective against *E. coli* O157:H7, *S. enterica* subsp. *enterica* Typhimurium, and *S. flexneri* LMG 10472 using spot overlay assay, with similar levels of inhibition as the model probiotics (*Lacticaseibacillus rhamnosus* GG and *Lactiplantibacillus plantarum* WCFS1) used as controls. In the radial diffusion assay, CFS of all the LAB isolates displayed inhibitory activity against *E. coli* O157:H7 and *S. enterica* subsp. *enterica* Typhimurium. Eight LAB isolates tested showed inhibitory activity against *S. flexneri* LMG 10472 using radial diffusion assay (Table 2). All LAB isolates except for *S. thermophilus* 15E were effective against *L. monocytogenes* MB2022 using spot overlay assay with similar levels to that of the positive control and model probiotics, while nine of the isolates tested were also effective in the radial diffusion method.

**Table 2.** Antagonistic activity of the selected potential probiotic LAB strains by spot overlay and radial diffusion methods against 9 strains indicator foodborne pathogens.

| | Zone of Inhibition (mm) [1], Data Are Mean Values ± SD, (*n* = 3) | | | | Zone of Inhibition (mm)[2], Data Are Mean Values ± SD, (*n* = 3) | | | | | | | | | |
|---|---|---|---|---|---|---|---|---|---|---|---|---|---|---|
| | *L. monocytogenes* ATCC 19115 | *S. aureus* ATCC 25923 | *E. coli* ATCC 25922 | methicillin-resistant *S. aureus* | *E. coli* O157:H7 BRMSID188 | | *S. enterica* subsp. *enterica* var. Typhimurium NTCT 13347 | | *S. flexneri* LMG 10472 | | *L. monocytogenes* MB2022 | | *S. aureus* MI/1310/1938 | |
| LAB strain (Source) | Spot overlay | Spot overlay | Spot overlay | Spot overlay | Radial diffusion | Spot overlay | Radial diffusion | Spot overlay | Radial diffusion | Spot overlay | Radial diffusion | Spot overlay | Radial diffusion | Spot overlay |
| *Limosilactobacillus fermentum* 12A (1) | +++ | + | +++ | +++ | ++ | − | ++ | − | + | − | + | ++ | − | + |
| *Streptococcus thermophilus* 15E (1) | ++ | + | ++ | − | ++ | − | ++ | − | − | − | − | − | − | − |
| *L. fermentum* 25A (2) | ++ | ++ | +++ | ++ | ++ | − | ++ | − | + | − | ++ | ++ | − | ++ |
| *Lactiplantibacillus plantarum* 54B (5) | ++ | + | ++ | ++ | ++ | ++ | ++ | ++ | +++ | ++ | +++ | +++ | − | +++ |
| *L. plantarum* 54C (5) | ++ | + | ++ | ++ | ++ | ++ | ++ | ++ | ++ | +++ | ++ | +++ | − | ++ |
| *L. plantarum* 55A (5) | ++ | ++ | ++ | ++ | +++ | ++ | ++ | ++ | +++ | ++ | +++ | +++ | − | ++ |
| *Lactiplantibacillus pentosus* 55B (5) | +++ | ++ | ++ | ++ | ++ | ++ | ++ | ++ | ++ | ++ | + | +++ | − | +++ |
| *L. fermentum* 55E (5) | ++ | ++ | ++ | +++ | ++ | − | ++ | − | − | − | ++ | ++ | − | + |
| *Weissella confusa* 93A (9) | − | + | ++ | ++ | ++ | ++ | ++ | ++ | − | ++ | − | +++ | − | ++ |
| *L. fermentum* 94E (9) | ++ | +++ | ++ | ++ | +++ | − | ++ | − | ++ | − | ++ | ++ | − | − |
| *Pediococcus pentosaceus* 95E (9) | +++ | ++ | +++ | ++ | ++ | ++ | ++ | ++ | ++ | +++ | ++ | +++ | − | ++ |
| Chlorhexidine 0.2% | ++ | + | + | + | | | | | | | | | | |
| *Lacticaseibacillus rhamnosus* GG | | | | | +++ | +++ | ++ | ++ | ++ | +++ | +++ | +++ | − | +++ |
| *L. plantarum* WCFS1 | | | | | +++ | ++ | ++ | ++ | ++ | +++ | +++ | ++ | − | ++ |
| Hexetidine 0.1% | | | | | ++ | ++ | + | − | ++ | + | +++ | +++ | +++ | +++ |

Chlorhexidine 0.2% and hexetidine 0.1% = Positive controls. Source: 1 = Commercially fermented, yogurt; 2 = Commercially fermented, cheese; 5 = Spontaneously fermented, cheese; 9 = Spontaneously fermented, cheese; 10 = Industrially fermented, probiotic yogurt. [1] Results of experiments of inhibition at AHRI: –no inhibition; low, + (9–14 mm); moderate, ++ (14–19 mm), and high inhibition, +++ (>19 mm). [2] Results of experiments of inhibition at LAMB: for radial diffusion assay: − = no inhibition; low, + (6–8 mm); moderate, ++ (8–11 mm), and high inhibition, +++ (>11 mm); for Spot assay: − = no inhibition; low, + (5–7 mm); moderate, ++ (7–10 mm); and high inhibition, +++ (>10 mm).

Nine of the eleven LAB isolates displayed activity against *S. aureus* MI/1310/1938 in the spot overlay method, but no CFS of the isolates tested (including model probiotics) could replicate the activity in the radial diffusion method (Table 2). Subsequently, the time-course effect of the 11 LAB isolates CFS on the growth of *S. aureus* MI/1310/1938 was measured in a more fine-scale, longitudinal liquid culture growth assay. Stronger longitudinal effects of LAB isolates CFS on the growth of *S. aureus* MI/1310/1938 were observed for four active LAB isolates (*Lactiplantibacillus plantarum* 54B, 54C, 55A, and *Lactiplantibacillus pentosus* 55B) compared to the model gastrointestinal probiotics (*Lacticaseibacillus rhamnosus* GG and *Lactiplantibacillus plantarum* WCFS1) (Figure 3A). *P. pentosaceus* 95E displayed significant inhibitory activity comparable to the model probiotics but lower than the four isolates (Figure 3A). The growth curve's area under the curve (AUC) estimates total bacterial growth as it correlates with both the growth rate and maximum density [45]. Consequently, AUC analysis of *S. aureus* MI/1310/1938 growth curves also revealed that five of the eleven LAB strains (*Lactiplantibacillus plantarum* 54B, 54C, 55A, *Lactiplantibacillus pentosus* 55B, and *P. pentosaceus* 95E) and model probiotics significantly inhibited growth ($p < 0.0001$) of *S. aureus* MI/1310/1938 compared to MRS medium control (Figure 3B). Although the differences in mean AUC of the LAB isolates 55E and 94E were statistically significant ($p < 0.05$) compared to that of the MRS medium control, these isolates were shown to be weak inhibitors, as they had overlapping growth curves with the medium (Figure 3A) and larger AUC values (Figure 3B). MRS broth (used as negative control) only induced a small delay in growth of the indicator pathogen.

To explore medium acidification as an antipathogenic mechanism of the LAB isolates, the CFS (Supplemental Table S3) was neutralized to pH 7.4, and subsequent radial diffusion assay against all indicator pathogens and longitudinal time-course analysis against *S. aureus* MI/1310/1938 were performed. The assays showed that antimicrobial activity of the CFS was pH-dependent, as the inhibition completely disappeared. Strong acidifiers (*Lactiplantibacillus plantarum* 54B, 54C, and 55A, *Lactiplantibacillus pentosus* 55B, *P. pentosaceus* 95E, *Lacticaseibacillus rhamnosus* GG, and *Lactiplantibacillus plantarum* WCFS1) with CFS pH < 4 also showed higher inhibition ($p < 0.05$) against the pathogenic strains tested.

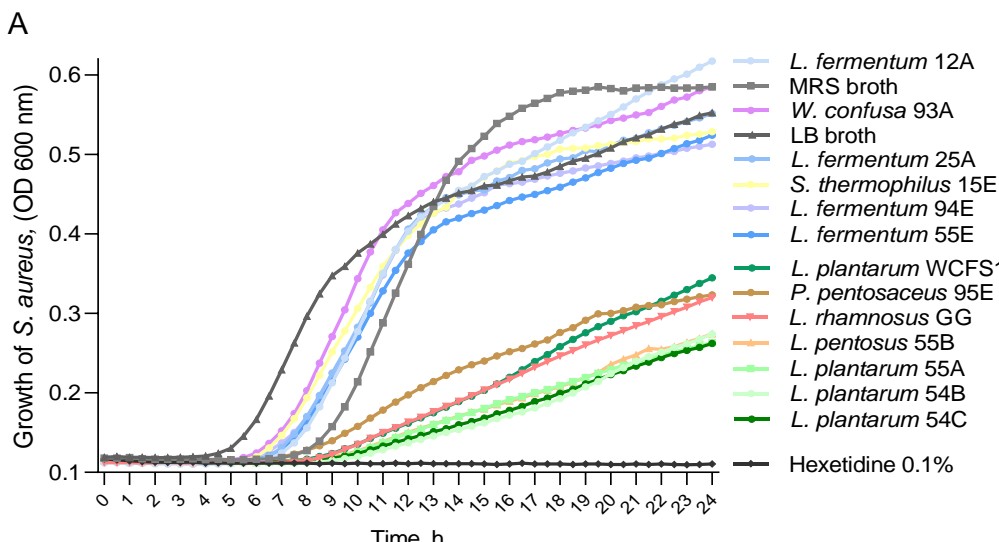

**Figure 3.** *Cont.*

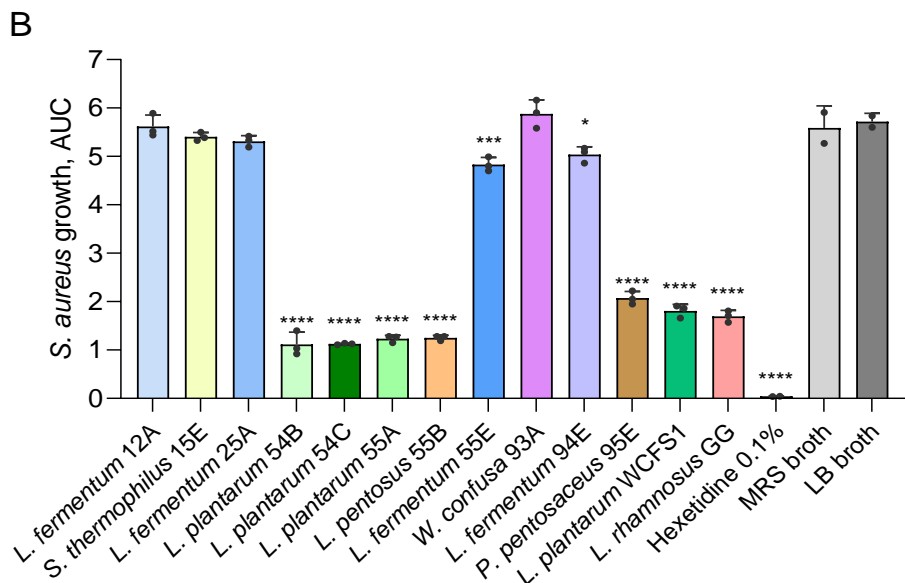

**Figure 3.** Effect of the LAB strains cell-free culture supernatant (CFS) against the growth of *S. aureus* MI/1310/1938 in LB broth: (**A**) Growth curves of *S. aureus* over the course of 24 h, non-inoculated MRS and LB broth and 0.1% hexetidine were used as negative and positive control, respectively. Curves of the most active four LAB strains (*L. plantarum* 54B, 54C, 55A, and *L. pentosus* 55B) are below the curves for the model probiotics (*L. rhamnosus* GG and *L. plantarum* WCFS1), indicating isolates were strong inhibitors. Curves for 95E and model probiotics are overlapping since 95E showed comparable inhibitory activity against the pathogen as model probiotics. (**B**) Area under the curve (AUC) of *S. aureus* growth curves. Bars depict AUC means ± SD per condition (*n* = 3). 55E and 94E have large AUC since they are weak inhibitors. * *p* < 0.05, *** *p* < 0.001, **** *p* < 0.0001 compared to *S. aureus* grown with MRS broth control. *L. plantarum*, *Lactiplantibacillus plantarum*; *L. pentosus*, *Lactiplantibacillus pentosus*; *L. fermentum*, *Limosilactobacillus fermentum*; *P. pentosaceus*, *Pediococcus pentosaceus*; *S. thermophilus*, *Streptococcus thermophilus*; *W. confusa*, *Weissella confusa*.

### 3.4. Selected Ethiopian Dairy LAB Isolates Activate NF-κB and IRF Pathways in Human Monocytes

Immunomodulation is one of the potential mechanisms of action of probiotics. In this study, the eleven selected LAB strains were further explored for their capacity to stimulate the NF-κB and IRF pathways as key for antipathogenic defenses in human monocytes. Nine out of the eleven tested LAB isolates significantly (*p* < 0.05) induced NF-κB, while *S. thermophilus* 15E and *W. confusa* 93A did not (Figure 4A). Of note, the tested LAB strains demonstrated variable strain-dependent immunostimulatory capacities. For example, while *Limosilactobacillus fermentum* 25A showed strong NF-κB activation, the other *Limosilactobacillus fermentum* strain 55E had a lower activity (Figure 4A). Three of the tested isolates, i.e., *Lactiplantibacillus plantarum* 54B and 55A and *P. pentosaceus* 95E, also displayed significant IRF induction, even higher than the model probiotic *Lactiplantibacillus plantarum* WCFS1 (Figure 4B). Several tested isolates demonstrated a trend towards IRF induction, including *Limosilactobacillus fermentum* 25A and 94E, *Lactiplantibacillus plantarum* 54C, and *Lactiplantibacillus pentosus* 55B, but this was not statistically significant in the tested conditions.

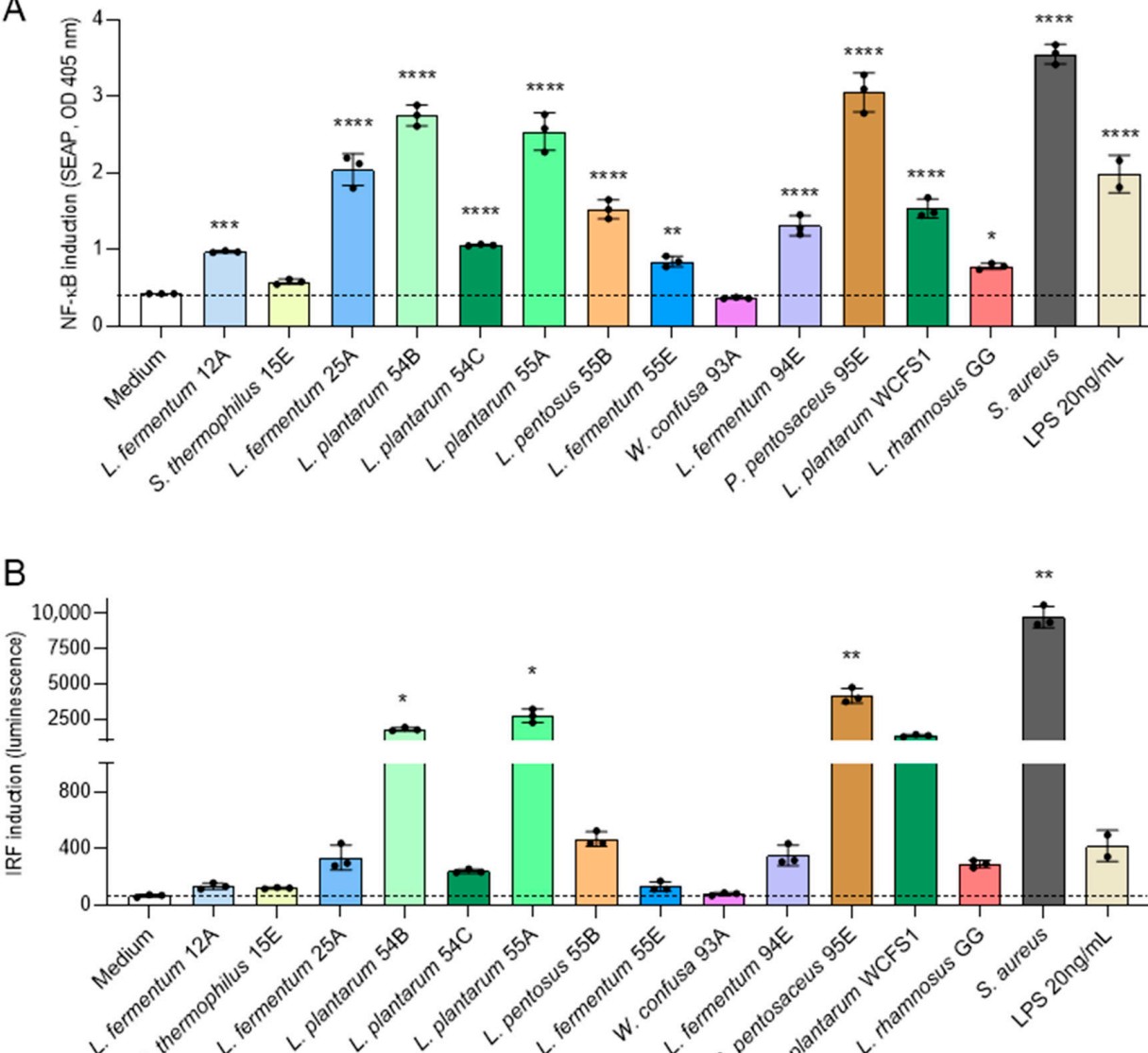

**Figure 4.** Immunostimulatory (**A**) NF-κB and (**B**) IRF activation by LAB strains in THP1-Dual human monocytes. *S. aureus* MI/1310/1938 was used as a Gram-positive pathogenic control strain. Bars depict mean ± SD per condition (*n* = 3). * $p < 0.05$, ** $p < 0.01$, *** $p < 0.001$, **** $p < 0.0001$ compared to medium control without bacteria (indicated by dotted line); *L. plantarum*, *Lactiplantibacillus plantarum*; *L. pentosus*, *Lactiplantibacillus pentosus*; *L. fermentum*, *Limosilactobacillus fermentum*; *P. pentosaceus*, *Pediococcus pentosaceus*; *S. thermophilus*, *Streptococcus thermophilus*; *W. confusa*, *Weissella confusa*.

### 3.5. Antibiotic Susceptibility Profile of Select LAB Isolates as Candidate Probiotic Strains

According to a 2002 report jointly released by the WHO and FAO of the United Nations, microbial strains to be used as probiotics should be safe in the host, with gene transfer of especially antibiotic resistance markers listed as one of the potential adverse events associated with probiotic use [46]. Therefore, it is important to verify that LAB strains to be consumed as a probiotic lack transferable antimicrobial resistance markers on mobile elements prior to considering them safe for human and animal consumption [46]. In the present study, antibacterial susceptibility profile of the 11 LAB isolates to 8 antibiotics recommended by EFSA [41] (ampicillin, chloramphenicol, clindamycin, erythromycin, gentamycin, kanamycin, streptomycin, and tetracycline) was examined (Table 3). All 11 LAB isolates tested showed sensitivity to ampicillin, erythromycin, clindamycin, and chloram-

phenicol at the respective reference concentration (Supplemental Table S4). Strain *Lactiplantibacillus plantarum* 55A was resistant to gentamycin. All LAB strains except *S. thermophilus* 15E and *Lactiplantibacillus pentosus* 55B showed resistance to kanamycin. LAB resistance to aminoglycosides such as kanamycin is considered to be natural [47,48] and, therefore, non-transmissible, so these strains could still be considered for further development.

**Table 3.** Antibiotic susceptibility profile of potential probiotic strains from dairy products.

| Isolate | Amp | Gent | Kana | Strep | Eryth | Clind | TTC | CAF |
|---|---|---|---|---|---|---|---|---|
| *L. fermentum* 12A | S | S | R | S | S | S | S | S |
| *S. thermophilus* 15E | S | S | S | S | S | S | S | S |
| *L. fermentum* 25A | S | S | R | S | S | S | S | S |
| *L. plantarum* 54B | S | S | R | n.r | S | S | S | S |
| *L. plantarum* 54C | S | S | R | n.r | S | S | S | S |
| *L. plantarum* 55A | S | R | R | n.r | S | S | S | S |
| *L. pentosus* 55B | S | S | S | n.r | S | S | S | S |
| *L. fermentum* 55E | S | S | R | S | S | S | S | S |
| *W. confusa* 93A | S | S | R | S | S | S | S | S |
| *L. fermentum* 94E | S | S | R | S | S | S | S | S |
| *P. pentosaceus* 95E | S | S | R | S | S | S | S | S |
| *L. rhamnosus* GG | S | R | R | S | S | S | S | S |
| *L. plantarum* WCFS1 | S | S | S | S | S | S | S | S |

(Amp: ampicillin; Gent: gentamycin; Kana: kanamycin; Strep: streptomycin; Eryth: erythromycin; Clind: clindamycin; TTC: tetracycline; CAF: chloramphenicol; n.r.: not required). *L. plantarum*, *Lactiplantibacillus plantarum*; *L. pentosus*, *Lactiplantibacillus pentosus*; *L. fermentum*, *Limosilactobacillus fermentum*; *P. pentosaceus*, *Pediococcus pentosaceus*; *S. thermophilus*, *Streptococcus thermophilus*; *W. confusa*, *Weissella confusa*.

## 4. Discussion

Although a large variety of spontaneously fermented foods exist in Ethiopia, their microbial constituents are largely underexplored. However, they form an interesting source of potentially novel isolates for applications in fermented foods and as probiotics. Isolating and characterizing LAB strains directly from widely consumed fermented foods is a particularly promising approach because of their applicability to fermented foods and their increased probability of being safe for oral consumption. In this work, we present one of the first dedicated studies on Ethiopian LAB strains isolated from different dairy sources, evaluating their efficacy and antibiotic susceptibility profile as potential probiotics.

A total of 27 LAB isolates were identified from Ethiopian yogurt and cheeses with *16S rRNA* gene Sanger sequencing: *Limosilactobacillus* (19), *Lactiplantibacillus* (4), *Streptococcus* (1), *Enterococcus* (1), *Pediococcus* (1), and *Weissella* (1) spp. The presence of these genera is consistent with Girma et al. [49], who isolated LAB (*Lactobacillus* (current reclassification as *Lactobacillus*, *Lacticaseibacillus*, and *Lactiplantibacillus* [9]), *Lactococcus*, *Leuconostoc*, *Pediococcus*, *Streptococcus*, *Enterococcus* spp.) from other fermented Ethiopian traditional dairy products (Ergo, Ayib, and Metata Ayib). Colombo et al. [49] also reported that *Lactobacillus* (current reclassification as *Lactobacillus*, *Lacticaseibacillus*, *Lactiplantibacillus*, and *Schleiferilactobacillus* [9]), *Pediococcus* spp., and *Weissella paramesenteroides* were the species isolated from a Brazilian dairy production environment. *Limosilactobacillus fermentum* was the predominant (70.4%) species in our samples, and this is in line with the report of Taye et al. [50] from cow milk and milk products from Ethiopia. However, the fact that *Limosilactobacillus* is isolated so often in fermented Ethiopian dairy products is of particular interest and extends the habitats of this genus because it seems to be different from other

geographical regions, where *Limosilactobacillus* is not often linked to dairy but rather to chicken and animal hosts [51].

Survival in the GI tract is a desirable property required for probiotics intended for oral administration. The tolerance of our LAB isolates to bile salts and acidic pH was studied in vitro to predict bacterial survival after oral administration. The acidic and protease-rich conditions of the stomach are generally the strongest barrier for probiotics [52]. The LAB isolates showed resistance to 4 h exposure to 0.5% bile salt and 3 h exposure to pH 3, with bile salt tolerance being universal, indicating good candidates as gastrointestinal probiotics.

Probiotics can exert their beneficial properties through many different mechanisms [46]. One of the potential probiotic properties of strains is antimicrobial activity. Ethiopia has a large burden of foodborne diseases [17,19], for which probiotics could be a good alternative to traditional antibiotic treatment. In the present study, three approaches were utilized to assess antipathogenic activity: radial diffusion, a spot overlay assay, and antimicrobial activity screening of CFS in liquid culture assays for the main causes of infection in Ethiopia (*E. coli*, *S. enterica* subsp. *enterica* var. Typhimurium, *S. aureus* (including MRSA) and *S. flexneri* and *L. monocytogenes*). In the radial diffusion assay, all CFS of the tested LAB isolates—containing secreted metabolites—displayed inhibition activities against *E. coli* O157:H7 and *S. enterica* subsp. *enterica* Typhimurium, while, because of the specificity of the spot overlay tests [53], only six of the eleven strains tested were effective against these two pathogens using a spot overlay assay that monitors more the live interaction between pathogen and potential probiotics (Table 2). To further explore and confirm the antimicrobial activity of the CFS of LAB isolates against *S. aureus* MI/1310/1938, we performed an inhibition experiment with the CFS and monitored the growth of *S. aureus* MI/1310/1938 for 24 h. A confirmed inhibitory activity was recorded for the five of eleven LAB isolates CFS (Figure 3). Three of the five isolates that showed antimicrobial activity against all nine strains of indicator pathogens using all methods and protocols tested belonged to the genus *Lactiplantibacillus plantarum*. Al-Madboly and Abdullah [54] detected and reported five potent antibacterial *Lactiplantibacillus plantarum* isolates recovered from fermented milk samples in Egypt, which were able to inhibit all the eight tested pathogenic bacterial strains from five pathogenic species (*S. aureus*, *E. faecalis*, *E. coli*, *S. flexneri*, and *S. enterica* subsp. enterica serovar *Typhi*). The LAB CFS neutralized to pH 7.4 failed to show any antagonistic activity, indicating that antimicrobial activity of the isolates is probably mainly due to the production of acidic substances. Similarly, Van den Broek et al. [40], Spacova et al. [55], and Reuben et al. [53] reported a loss of antagonistic activity by most LAB CFS tested against selected pathogens after neutralizing the supernatant, but these previous studies did not use native Ethiopian isolates.

In addition to their antipathogenic and adaptation properties, probiotics capable of modulating the immune system are highly promising for application against diseases related to immune imbalances, such as allergic diseases [56], inflammatory bowel disease [57], and even COVID-19 [58]. Our results demonstrate that nine of the eleven tested LAB isolates from Ethiopian fermented dairy products were capable of activating the key immune transcription factor NF-κB to similar levels as the model probiotic strain, *Lactiplantibacillus plantarum* WCFS1 [44]. The latter strain was recently successfully implemented as part of a throat spray in COVID-19 patients [58]. NF-κB activation by LAB could help stimulate antipathogenic immune responses and correct the development and regulation of immune self-tolerance [59–62]. Furthermore, our selected LAB isolates demonstrated activation of IRF. IRF is especially necessary for host antiviral defenses. For example, activation of IRF by *Lactobacillus acidophilus* [63] or dsDNA of various LAB [31] has previously been linked to protective IFN-β response induction in host cells. Importantly, we observed that the immunostimulatory activity of LAB was strain-specific. This supports previous results on LAB that immunostimulatory activity is strain-specific [32] and highlights the need to select appropriate probiotic strains for each envisioned application. Three of the eleven tested strains belonging to *Lactiplantibacillus plantarum* (54B, 54C, and 55A), *Lactiplantibacillus pentosus* 55B, and *P. pentosaceus* 95E demonstrated the most efficient NF-κB and IRF activation

similar or higher than the model probiotic *Lactiplantibacillus plantarum* WCFS1, suggesting these strains are promising candidates to induce protective immune responses in the host. This might be especially promising if these strains are used in fermented foods. Of note, a recent systematic review and meta-analysis focusing on the effects of orally administered probiotics on respiratory tract infections in adults specifically demonstrated that infection duration was more efficiently reduced when fermented dairy was used as the delivery matrix for probiotics [64].

To assess the prospective application of the selected LAB strains as probiotics or in food/feed, we next considered the recommendations by EFSA [41] regarding antibiotic resistance. LAB can serve as a reservoir for antibiotic-resistant genes and transfer them to other microorganisms, including pathogens [65]. A probiotic candidate should be verified for lack of acquired transferrable resistances. Therefore, susceptibility to the recommended antibiotics should be assessed for all potential probiotic strains [41]. LAB resistance to aminoglycosides (gentamycin, kanamycin, streptomycin, or neomycin) and glycopeptide (vancomycin), in most cases, is considered to be natural and, therefore, non-transmissible [47,48,53]. Hence, all tested LAB isolates are presumed to be safe regarding antibiotic resistance. Although LAB strain *P. pentosaceus* 95E had a lower survival rate at low pH, it is one of the best performers in antagonistic activity and immunostimulatory assays. As there is no clear cut-off value for in vitro GI conditions resistance and proof of benefit can be established in further in vivo and human studies, it can be taken as a promising probiotic candidate. Overall, we demonstrated that five (*Lactiplantibacillus plantarum* 54B, 54C, and 55A, *Lactiplantibacillus pentosus* 55B, and *P. pentosaceus* 95E) select LAB isolates have promising antimicrobial and immunostimulatory properties and are presumed to be safe with respect to antibiotic resistance (Table 4) and could, thus, be considered as promising candidates for use in fermented foods or as food supplements.

**Table 4.** Summary of results of probiotic properties of LAB strains to select as candidate probiotics.

| Property Tested | | Good Candidate LAB Strains | | | | | | Poor candidate LAB Strains | | | | |
|---|---|---|---|---|---|---|---|---|---|---|---|---|
| | | *L. plantarum* 54B | *L. plantarum* 54C | *L. plantarum* 55A | *L. pentosus* 55B | *P. pentosaceus* 95E | *W. confusa* 93A | *L. fermentum* 12A | *S. thermophilus* 15E | *L. fermentum* 25A | *L. fermentum* 55E | *L. fermentum* 94E |
| Antipathogenic activity against | *L. monocytogenes* ATCC 19115 | √ | √ | √ | √ | √ | − | √ | √ | √ | √ | √ |
| | *S. aureus* ATCC 25923 | √ | √ | √ | √ | √ | √ | √ | √ | √ | √ | √ |
| | *E. coli* ATCC 25922 | √ | √ | √ | √ | √ | √ | √ | √ | √ | √ | √ |
| | Methicillin-resistant *S. aureus* (MRSA) | √ | √ | √ | √ | √ | √ | √ | − | √ | √ | √ |
| | *L. monocytogenes* MB2022 | √ | √ | √ | √ | √ | − | √ | − | √ | √ | √ |
| | *S. enterica* subsp. *enterica* var. Typhimurium NTCT 13347 | √ | √ | √ | √ | √ | √ | − | − | − | − | − |
| | *E. coli* O157:H7 BRMSID188 | √ | √ | √ | √ | √ | √ | − | − | − | − | − |
| | *S. aureus* MI/1310/1938 | √ | √ | √ | √ | √ | − | − | − | − | − | − |
| | *S. flexneri* LMG 10472 | √ | √ | √ | √ | √ | − | − | − | − | − | − |
| In vitro GI conditions resistance | pH= 3 | √ | √ | √ | √ | √ | − | √ | √ | √ | √ | √ |
| | Bile salt 0.5% | √ | √ | √ | √ | √ | √ | √ | √ | √ | √ | √ |
| NF-κB activation | | √ | √ | √ | √ | √ | − | √ | − | √ | √ | √ |
| IRF induction | | √ | ns | ns | ns | √ | − | − | − | ns | − | ns |
| AST | | √ | √ | √ | √ | √ | √ | √ | √ | √ | √ | √ |

√ = robust/significant/safe; AST = antibiotic susceptibility test; − = no activity.

## 5. Conclusions

In this study, five LAB isolates from traditional cottage cheese showed in vitro broad-spectrum antimicrobial activities against nine strains of foodborne pathogens from five species and stimulated key immune pathways in human cells. All five LAB isolates complied with antibiotic resistance recommendations. These findings indicate that the selected LAB strains are promising probiotic candidates for use in fermented foods and food supplements and highlight the potential of traditional fermented dairy products as a source of novel probiotic bacteria. They can be considered probiotic strains once their health benefits are documented in a clinical trial as a next step.

**Supplementary Materials:** The following supporting information can be downloaded at: https://www.mdpi.com/article/10.3390/fermentation9030258/s1, Table S1: LAB load of dairy samples; Table S2: Antagonistic activity of the selected 43 potential probiotic LAB strains by spot overlay method against; *L. monocytogenes* (ATCC 19115), *S. aureus* (ATCC 25923), *E. coli* (ATCC 25922), methicillin resistant *S. aureus* (MRSA); Table S3: pH of the corresponding LAB isolates cell-free culture supernatants; Table S4: Concentration of antibiotics used for determination of antibacterial susceptibility test.

**Author Contributions:** Concept: S.G., S.H.M. and E.E.; Experimental design: S.G., W.V.B., S.L. and I.S.; Experimental work: S.G., W.V.B., M.D. and A.A. Data analysis: S.G., W.V.B. and I.S.; Writing—original draft: S.G.; Writing—review and editing: W.V.B., S.L., I.S., M.D., W.M.W., S.H.M. and E.E. All authors have read and agreed to the published version of the manuscript.

**Funding:** This Research received no external funding.

**Institutional Review Board Statement:** This study was approved by Ethics Committee of the School of Pharmacy, College of Health Sciences, Addis Ababa University with reference number ERB/SOP/15/10/2018.

**Informed Consent Statement:** As the study does not involve human subjects, obtaining informed consent is not required.

**Data Availability Statement:** All pertinent data are included in the article. Other data can be obtained from the corresponding authors on reasonable request.

**Acknowledgments:** We acknowledge Arba Minch University and Addis Ababa University for their financial support in this study. We would like to thank the Armauer Hansen Research Institute for granting SG access to its laboratory and the Laboratory of Applied Microbiology and Biotechnology, University of Antwerp, for granting SG access to its laboratory and generously providing all materials needed to conduct the microbiological, molecular, and cell culture experiments. We would also like to thank the Research staff of the laboratories.

**Conflicts of Interest:** The authors declare that they have no competing interests related to this work. S.L. is an academic board member of the International Scientific Association on Probiotics and Prebiotics (ISAPP) and co-founder of YUN. However, these organizations were not involved in this work.

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
