# Peer review of "Antibacterial and Immunostimulatory Activity of Potential Probiotic Lactic Acid Bacteria Isolated from Ethiopian Fermented Dairy Products"

_fermentation, doi:10.3390/fermentation9030258_

Round 1

Reviewer 1 Report

The authors report a multi-pronged study to identify and characterize candidate probiotics from ethiopian fermented foods. The reviewer is quite pleased with the quality of writing, data presentation, and conclusions drawn. Minor comments are provided below.

L71: … such as those documented for…

L122: … and one cheese product from two…

L141: … by gram-reaction…

L214: Revise to report centrifuge in x g units if known/can be calculated.

Figure 3: The B component does not appear to add much to the growth curves depicted in the A panel. The growth curve demonstrates inhibition, time to log phase entry, or replication kinetics. The AUC showing B panel does not. Recommend removing it and either providing as supplement figure or discarding.

Table 4: Reviewer recommends including the tested concentration of each antibiotic to allow deeper interpretation of the susceptibility/resistance capacities of tested microbes.

Author Response

Thank you for your constructive criticisms. Please find below a point-by-point response for your queries.

  • L71: … such as those documented for…
  • Accommodated.
  • L122: … and one cheese product from two…
  • Accommodated.
  • L141: … by gram-reaction…
  • Modified in a better way.
  • L214: Revise to report centrifuge in x g units if known/can be calculated.
  • It is described as the relative centrifugal force “g”.
  • Figure 3: The B component does not appear to add much to the growth curves depicted in the A panel. The growth curve demonstrates inhibition, time to log phase entry, or replication kinetics. The AUC showing B panel does not. Recommend removing it and either providing as supplement figure or discarding.
  • We appreciate the comments of the reviewer. However, we would like to emphasize that the area under the curve is a useful metric to summarize a growth curve, as it integrates the different growth parameters into a single value, and allows straightforward statistical analysis between conditions. Furthermore, it is more visually clear when many different conditions are tested. For these reasons, we prefer to keep Figure 3B, as it clearly (quantitatively speaking) shows the difference in activity between the isolates and the medium.
  • Table 4: Reviewer recommends including the tested concentration of each antibiotic to allow deeper interpretation of the susceptibility/resistance capacities of tested microbes.
  • We used the concentration of antibiotics recommended by the European Food safety Authority (EFSA) and cited it in the Method section (Reference no. 41). It is not possible to show the concentration of each antibiotic used in Table 4, as it varies with species. As both Reviewers suggested the inclusion of antibiotic concentrations, we have included the concentrations used in this study as Supplemental Table S4.

Author Response

Dear Reviewer 2,

Thank you for your constructive criticisms. Please find below a point-by-point response for your queries.

The article deals with the screening and proving probiotic properties of LAB isolates originated from Ethiopian traditional fermented food products. Authors focused on identification of isolates, their antimicrobial and immunomodulatory properties, and on resistance of isolates against antibiotics. The manuscript presents interesting and promising results in the area of probiotic bacteria and their usage; however, some parts of the manuscript need additional input from the authors.

  Specific comments

  • In lines 57-60 you are specifying LAB. According to the Zheng et al. (2020) the genera Lactobacillus is now divided into 23 genera including Lactiplantibacillus, Lacticaseibacillus, Limosilactobacillus, etc.; however, in Rezac et al. 2018 these genera together with Pediococcus and Weisella as LAB were not mentioned. Moreover, LAB group includes besides Streptococcus, Pediococcus, Leuconostoc, Weisella and former Lactobacillus, also other genera. Please edit these sentences.
  • In lines 58-60, we’ve edited the genus Lactobacillus as ‘the emended genus Lactobacillus’ and changed the ref. Rezac et al. 2018 to Hutkins, 2019, in which the specified genera are mentioned.
  • In lines 79-85 you mentioned that majority of foodborne diseases in Africa and in Ethiopia are caused by Salmonella, coli, L. monocytogenes, Campylobacter and MRSA. However, your tested bacteria were Salmonella, Shigella, E. coli, L. monocytogenes and MRSA. Can you explain why you exchanged Campylobacter by Shigella?
  • It would be beneficial to describe also the process of fermentation of yogurt and cheese, not only of cottage cheese. Also, why you used only 1 sample of yogurt or cheeses, are these specific samples unique in some point of view?
  • Fermentation processes of yoghurt and cheese are described (see lines 355-358).
  • We used one sample each from commercial dairy products, because the study was a type of exploratory research and we wanted to mainly focus on the traditional than the commercial products, as the former are less explored than the latter.
  • To isolate microorganisms from solid sample, 10g is not as suitable as e.g., 45g or more.
  • Although we agree that a larger sample size could be used, 10 g has also been demonstrated in various studies to provide as good yield as one can get from larger sample size (Reuben et al., 2020; Silva, 2013). Additionally, the microbial load of our fermented food was high, as 4th – 5th dilutions were needed to obtain countable plates.
  • You mentioned that you identified only 5 random colonies grown on plates with 30-300 colonies (in selective or elective media 15-150 is enough) and then you used % of confirmed colonies to calculate the CFU/ml of LAB. This approach is at least very strange, as it can be sometimes the question of lucky/unlucky choice in case of 5 random colonies.
  • We used 30-300 colonies as it is a common practice (O’Toole, 2016). As regards to the CFU calculation, ideally, bacterial count can be done by total 16S rRNA amplicon sequencing (or qPCR). However, rough estimation using 5 random pure colonies from each dilution is used in wet labs (Silva, 2013). Previously, all isolates grown on MRS agar were counted and reported as LAB (Abegaz, 2007).
  • During mimicking the GI tract conditions for survival of LAB you used aerobic conditions instead of anaerobic that are in GIT. Can you explain it?
  • It is well known that most of the GI tract is anaerobic. The term ‘aerobic” was inserted in the manuscript to show that anaerobic jar was not used in the experiment. We would also like to point out that we focused on the acidic conditions of the stomach, which is known to not be 100% anaerobic, but rather aerobic/microaerophilic (Mettu et al., 2021). Thus, what we did was pouring the solution into a cryotube and incubating by using a closed incubator with stirrer. Most studies simulate the GI condition in such a manner. We removed the word “aerobic” to avoid confusion.
  • The detection limit of 103 CFU/ml is very high for plate count dilution method, usually it is 101 CFU/ml. Can you explain it?
  • We used 103 CFU/ml as a detection limit because the volume applied to the circular plate was 10 microliters, as we counted 30 colonies as minimum amount according to the protocol. Had we applied 1 ml, 101 would have been the detection limit.
  • I´m little bit confused by the procedure of spot overlay assay (chapter 2.4.1). Firstly, what is the difference between spot overlay and agar diffusion assays? Secondly, the spots of LAB on MRS agar were made at AHRI and the spots of pathogens on MHA and LB agars were made at LAMB? And then you overlayed the agar containing the pathogens over the spot inoculated plates? Containing what?
  • We thank you for raising these questions. We have included a detailed description below and elaborated the details in the manuscript (lines 202-206) to improve clarity.
  • Spot overlay assay,

LAB isolates were grown in liquid media (overnight at 37C). 2ul of the grown culture was spotted on dedicated agar medium (MRS, MHA or LB agar) and incubated for 24-48 h to allow LAB growth as spot. The indicator pathogen is grown in liquid media overnight and then mixed with soft agar (0.5% agar) and uniformly poured over the plate containing spot (spot is overlaid), incubated at 37C. After 24 h the diameter of the inhibition zone (defined as zone without visual growth of the pathogen) around the spot is measured.

  • Agar diffusion assay (we did not do this assay but presented it for clarity)

The agar plate is swabbed with the test organism, and then paper disks containing a defined antibiotic concentration are placed on the lawn of bacteria, incubated overnight (24 h) and clear zone diameter around the disks is measured. In spot overlay assay, LAB are spotted over agar and allowed to grow first, and then pathogens in soft agar are overlaid. In agar diffusion, pathogens are swabbed on the agar first and then disks are placed on it.

  • Radial diffusion assay

Compared to the spot overlay assay (where living bacterial cells/colonies are used), in the radial diffusion assay the supernatant of the overnight LAB culture is used. In short, the LAB isolates are grown overnight at 37C. These grown liquid cultures are then centrifuged at 2484 g for 15 min at 4C and filter sterilized to make sure supernatant only contains metabolites produced by the LAB and no living bacterial cells. Indicator pathogen is grown overnight in liquid culture and then mixed with agar medium and poured in the petri dish. Using sterile glass Pasteur pipette, 6mm diameter wells are punched in the solidified agar (which contains our indicator pathogen). Sterile LAB supernatant is dispensed within the wells and plates are incubated at 37C. After 24-48h inhibition zones were measured.

Both spot and radial diffusion tests examine the antimicrobial potential for the selected LAB isolates but in the radial diffusion assay, the antimicrobial effect of only the metabolites produced are studied, whereas in the spot overlay method direct cell interactions are also included.

  • We overlaid the soft agar containing the indicator pathogens over the spot inoculated plates, containing spot (acting as a disk) of grown LAB.
  • Why you tested the antibacterial activity of CFS in liquid culture only in case of aureus (chapter 2.4.3)?
  • As it is indicated in the manuscript (lines 353-357), nine of the eleven LAB isolates displayed activity against aureus MI/1310/1938 in the spot overlay assay. However, no CFS of the isolates tested (including model probiotics) were able to replicate the activity in the radial diffusion method. However, this method is a single time point measurement after 48 h whereas the effect on S. aureus could be more profound using a longitudinal approach (with measurements at shorter intervals). Therefore, we opted to check whether the CFS of LAB were really inactive against S. aureus. To this end, the time-course effect of the 11 LAB isolates CFS on the growth of S. aureus MI/1310/1938 was measured in a more fine-scale, longitudinal liquid culture growth assay.
  • It is not clear what concentration of antibacterial solution was used in chapter 2.6. Moreover, instead of visible growth observation, the OD of grown isolates at specific conditions would be more precise.
  • This issue was raised by both reviewers and we have addressed it above in our reply to Reviewer 1. Although the EFSA recommendation is a public knowledge, we have included the concentrations used as Supplementary Table S4.
  • References for the protocol we used (CLSI, 2012; EFSA, 2012) to test susceptibility of LAB for the antibiotics require to read results, as completely inhibited growth in wells, with the unaided eye. We also measured OD and we did not find any difference with the visual observation.
  • According to Fig. 1, you were supposed to analyse 43 putative LAB isolates via 16S rRNA gene sequencing; however, in Tab. 1 there are only 27 isolates. Can you add the rest of them?
  • 43 putative LAB isolates were found based on being gram-positive, catalase negative and non-motile. Out of these 43 isolates, 27 were selected based on preliminary screening of antibacterial activity performed at AHRI using the spot overlay assay (see the revised Supplementary Table S3) as well as to cover diversity of sample origins for 16S rRNA gene sequencing to identify their species. These 27 were then used in the screening pipeline (Fig. 1) to select potential probiotic strains. Additional clarification on how the isolates were selected was added within the text (lines 361-363) and within the legend of Figure 1.
  • Based on Tab. 1, can you specify the key for choosing specifically those 11 isolates, e.g., why from yoghurt only 1 Limosilactobacillus fermentum (with similarity 99.92% not another with 100%) and why from spontaneously fermented cheeses 8 isolates? Probably it is due to the resistance to GI tract conditions, but it is not obvious from the table.
  • If the pair-wise similarity is more than 98.7%, species identification is assumed based on 16S rRNA sequencing. This could not be therefore used as a criterion for selection of isolates for further activity assessment. Thus, we largely based our selection on activity displayed by the isolate.
  • We felt that Limosilactobacillus fermentum isolates from commercial yoghurt (10 out of 11 were L. fermentum) might be of the same strain and chose just 1 that performed well on the spot overlay assay. Isolates from spontaneously fermented cheeses which performed good on the spot overlay assay were of different species or different strains of the same species (L. plantarum).
  • Line 314: is the survival of isolate 95E in 50% of initial population enough to declare “that it exhibited resistance to low pH”? In addition, also the bars for sd are missing in this case. Moreover, you selected this isolate as promising probiotic, but the survival rate is lower than in case of other isolates.
  • Of course, isolate 95E had lower survival rate at low pH. However, it is one of the best performers in antagonistic activity and immunostimulatory assays. As there is no clear cut-off value for in vitro GI conditions resistance, proof of benefit can be established in further in vivo and human studies. Based on these arguments, we still take it as a promising candidate probiotic. Explanation on this also included in the manuscript lines 750-754.
  • The value measured for the three replicates of this isolate is the same and the SD will obviously be zero.
  • Line 323-331: Why you tested antagonistic activity of 27 isolates against only 4 pathogens? Why you excluded Shigella as you stated in chapter 2.4.? Based on Fig. 1 it seems, that you would have tested antimicrobial activity all 27 isolates against 5 pathogens to select 11 further candidates.
  • We followed activity guided selection and screening of isolates for probiotic properties. We conducted antagonistic activity for 27 LAB isolates against four indicator pathogens using spot overlay assay at AHRI. Based on this initial screening and taxonomic classification of the isolates (16S rRNA gene Sanger sequencing – species/strain variety) 11 LAB isolates were select for further detailed probiotic characterization, including additional in-depth antagonistic activity against 5 different pathogenic strains using both spot overlay and radial diffusion assay as well as longitudinal liquid culture growth assay against aureus MI/1310/1938. Subjecting the isolates to different assays increases robustness and reproducibility of the antagonistic activity.
  • It would be beneficial to have extended Table 5, not only for 5 promising isolates and confuse but for all other 11 isolates chosen for further screening.
  • We provided the data in the revised version of Table 5.
  • Why you screened resistance also against gentamycin, kanamycin, streptomycin, when as you mentioned in lines 533-534: “LAB resistance to aminoglycosides (gentamycin, kanamycin, streptomycin or neomycin) and glycopeptide (vancomycin), in most of the cases, is considered to be natural and, therefore, non-transmissible”?
  • Screening resistance of LAB isolates against aminoglycosides was performed based on European Food Safety Authority’s recommendation. This phenotypic AST is used only as a preliminary screening criterion to select isolates as candidate probiotics. Definitive test for resistance as well as virulence factors is whole genome sequence analysis as recommended by another EFSA statement (EFSA, 2021), that we plan to do it for our promising candidates. We included isolates resistant to aminoglycosides, as recommended by EFSA, in our study until the resistance is confirmed by whole genome sequencing.

Minor comments:

  • Writing of abbreviation for species, spp. is not in italics; however, writing of gene names, e.g., stx is with italic.
  • Accommodated.
  • You are mixing using mL and ml.
  • Accommodated.
  • The SI unit abbreviation for gram is “g” not “gm”.
  • Accommodated.
  • There are some grammar mistakes, e.g., wrong using/not using of define/indefinite article, using of prepositions, etc.
  • Checked and corrected.
  • It is confusing to use the same abbreviation for different Lactobacillaceae, e.g.: Lactiplantibacillus plantarum and pentosus, Limosilactobacillus fermentum, Lacticaseibacillus rhamnosus, etc.
  • Thanks for noticing it. We used the full name in cases where the first letter represents different genus and an abbreviation where it represents the same genus.

References

Abegaz, K., 2007. Isolation, characterization and identification of lactic acid bacteria involved in traditional fermentation of borde, an Ethiopian cereal beverage. Afr. J. Biotechnol. 6, 1469–1478.

CLSI (Ed.), 2012. Methods for dilution antimicrobial susceptibility tests for bacteria that grow aerobically: approved standard - ninth edition, Clinical and Laboratory Standards Institute. CLSI, Wayne, Pa.

EFSA, 2021. EFSA statement on the requirements for whole genome sequence analysis of microorganisms intentionally used in the food chain. EFSA J. 19. https://doi.org/10.2903/j.efsa.2021.6506

EFSA, 2012. Guidance on the assessment of bacterial susceptibility to antimicrobials of human and veterinary importance                EFSA Panel on Additives and Products or Substances used in Animal Feed (FEEDAP). EFSA J. 10. https://doi.org/10.2903/j.efsa.2012.2740

O’Toole, G.A., 2016. Classic Spotlight: Plate Counting You Can Count On. J. Bacteriol. 198, 3127–3127. https://doi.org/10.1128/JB.00711-16

Reuben, R.C., Roy, P.C., Sarkar, S.L., Rubayet Ul Alam, A.S.M., Jahid, I.K., 2020. Characterization and evaluation of lactic acid bacteria from indigenous raw milk for potential probiotic properties. J. Dairy Sci. 103, 1223–1237. https://doi.org/10.3168/jds.2019-17092

Silva, N. da, 2013. Microbiological examination methods of food and water: a laboratory manual, Second edition. ed. CRC Press, Taylor & Francis Group, Boca Raton.

Round 2

Reviewer 2 Report

Thank you very much for your responses that helped me to better understand your work, and I´m also thankful for your additional work that impoved quality of the manuscript. Good luck in your further research and publications.